# Large genomic deletions delineate *Mycobacterium tuberculosis* L4 sublineages in South American countries

**Andres Baena**[1,2,3], **Felipe Cabarcas**[4], **Juan C. Ocampo**[1,3], **Luis F. Barrera**[1,3,5], **Juan F. Alzate**[2,3,4] *

1 Grupo de Inmunología Celular e Inmunogenética (GICIG), Facultad de Medicina, Universidad de Antioquia, Medellín, Colombia, 2 Departamento de Microbiología y Parasitología, Facultad de Medicina, Universidad de Antioquia, Medellín, Colombia, 3 Sede de Investigación Universitaria-SIU, Universidad de Antioquia, Medellín, Colombia, 4 Centro Nacional de Secuenciación Genómica—CNSG, Universidad de Antioquia, Facultad de Medicina, Universidad de Antioquia, Medellín, Colombia, 5 Instituto de Investigaciones médicas, Universidad de Antioquia, Medellín, Colombia

* jfernando.alzate@udea.edu.co

**Data Availability Statement:** All newly generated genomic read data files are available from the NCBI SRA Database bioproject PRJNA867148.

**Funding:** Minciencias, Colombia. CODE: 111584467121, Contrato No. 393-2020.

## Abstract

*Mycobacterium tuberculosis* (Mtb) is still one of the primary pathogens of humans causing tuberculosis (TB) disease. *Mtb* embraces nine well-defined phylogenetic lineages with biological and geographical disparities. The lineage L4 is the most globally widespread of all lineages and was introduced to America with European colonization. Taking advantage of many genome projects available in public repositories, we undertook an evolutionary and comparative genomic analysis of 522 L4 Latin American *Mtb* genomes. Initially, we performed careful quality control of public read datasets and applied several thresholds to filter out low-quality data. Using a genome *de novo* assembly strategy and phylogenomic methods, we spotted novel south American clades that have not been revealed yet. Additionally, we describe genomic deletion profiles of these strains from an evolutionary perspective and report *Mycobacterium tuberculosis* L4 sublineages signature-like gene deletions, some of the novel. One is a specific deletion of 6.5 kbp that is only present in sublineage 4.1.2.1. This deletion affects a complex group of 10 genes with putative products annotated, among others, as a lipoprotein, transmembrane protein, and toxin/antitoxin system proteins. The second novel deletion spans for 4.9 kbp and specific of a particular clade of the 4.8 sublineage and affects 7 genes. The last novel deletion affects 4 genes, extends for 4.8 kbp., and is specific to some strains within the 4.1.2.1 sublineage that are present in Colombia, Peru and Brasil.

## Introduction

*Mycobacterium tuberculosis* (*Mtb*) is a major human pathogen, infecting 10 million people worldwide and killing 1.5 million in 2021 [1, 2]. *Mtb* belongs to the MTBC complex and is classified into nine main lineages (L1 to L9) [3, 4]. Specifically, lineage L4 is the most

**Competing interests:** The authors have declared that no competing interests exist.

geographically widespread of all lineages and is also the most prevalent in south America [4, 5]. Compared with other lineages, the L4 lineage has shown an increased virulence in *in vitro* infected macrophages and in animal models, although there is variability between different L4 *Mtb* strains [6]. In Latin America, the actual L4 *Mtb* sublineages have been determined mainly by European colonial movements, recent immigration, and population stratification [6].

Nowadays *Mtb* strain discrimination is performed using methods that analyze specific SNPs (barcoding) and whole genome sequencing (WGS) technologies. Although the SNP barcoding strategy do not offer the same resolution compared to WGS, they provide rapid and valuable insights into the population structure of circulating strains. The first well-known SNP barcode initiative used 60 loci. Later it was broaden to a 90 SNPs barcode panel that allows locating the *Mtb* strains within clades inside the seven human lineages and defined 86 sublineages [4].

As observed in other bacterial species, *Mtb* has evolved mainly through single nucleotide mutations and genome InDels [7]. Single nucleotide mutations are the primary source of variation in *Mtb* genomes, in addition to short indels (<50 bp) that can occur throughout the whole genome but are more commonly found in the PE-PPE genes [8–10]. On the other hand, large deletions, previously called large-sequence polymorphisms (LSPs), are also common and important because they usually lead to disruptions of CDSs but also can also work as junction points of truncated genes. This phenomenon has been shown to affect a range of metabolic pathways and putative virulence factors [10]. Some of these indels may cause attenuation like in the proved case of RD1 loss or may cause a hyper-virulent phenotype like in the kdpDE two-component system [11].

In this study, we assessed the quality of 866 publicly available genome projects (WGS) of *Mtb* L4 lineage, isolated in 9 different Latin-American countries. After discerning the evolutionary history of 522 isolates of the L4 sublineage in seven Central and South American countries. With this comprehensive phylogenetic analysis, we discovered modern country-specific lineages that thrive within the South American countries and that haven not been spotted so far. Thanks to the genome assembly strategy, we characterized large genomic deletion profiles of these strains, within an evolutionary framework and report signature-like gene deletions for certain South American L4 sublineages. We also present, to date, the most comprehensive analysis of large genomic deletions in the *Mtb* L4 lineage in Latin America.

## Materials and methods

### Bacterial culture and DNA extraction for the Colombian genomes

*Mycobacterium tuberculosis* was isolated from the sputum of HIV-negative recently diagnosed pulmonary TB patients (PTB, n = 88). These PTB patients were 53% women and 47% men between 25 to 35 years old. The decontaminated sputum was grown in Ogawa Kudoh medium and then transferred to 10mL of 7H9 liquid medium supplemented with OADC (10%), Tyloxapol (0.05%) and glycerol (0,05%). The samples were verified using the SD BIOLINE TB Ag MPT64 rapid test (Abbot, Illinois, USA). Then, the bacteria were cultured to an $OD_{600}$ nm of 0.5, harvested by centrifugation (3000rpm), and the pellet stored at -20°C. For genomic DNA extraction, we combined previously described protocols [12, 13].

The study procedure was approved by the Universidad de Antioquia human ethical committee, Medellin, Colombia. All methods were performed in accordance with the Declaration of Helsinki. All participants in the study were informed of the risk involved in the study, and voluntarily signed the written informed consent in order to have access to the sputum samples from each patient.

## Genome sequencing of Colombian genomes

The *Mycobacterium tuberculosis* WGS experiments were performed by Novogene (Sacramento, CA) using an Illumina Novaseq 6000 instrument. One Truseq nano DNA library prepared for each isolate and for each library we aimed for 1Gb raw bases reading PE reads of 150 bases. All sequenced libraries showed at least 90% of Q30 bases. Raw reads were deposited at the NCBI SRA database under the bioproject PRJNA867148 (https://dataview.ncbi.nlm.nih.gov/object/PRJNA867148?reviewer=okjkdkaevi75kfu8phla1a30su).

## Latin American *Mycobacterium tuberculosis* L4 genomic data downloaded from the NCBI SRA

We searched the NCBI SRA database for *M. tuberculosis* genomes projects sequenced using Illumina platforms (PE reads) that involved Latin American isolates and separated them according to the country of origin. We excluded all isolates labeled as *M. bovis* and that were classified into sublineages different than L4. For countries with hundreds or thousands of genomes, we selected those labeled as sublineage L4 and those with the raw read bases summed at least 500 Mb. If the list was still above 500 projects, we increased the threshold to a minimum of 900 Mb of raw reads. From this repository we selected the bioprojects PRJNA755956, PRJEB30933, PRJEB29069, PRJEB44165, PRJEB27366, PRJNA670836, PRJNA671770, PRJEB8689, PRJEB7669, PRJEB8689, PRJEB41837, PRJEB29408, PRJEB23681, PRJNA749058, PRJNA595834, PRJNA454477, PRJNA300846 and PRJNA422870.

As an additional control, we classified all the isolates using the *TB-pro*filer tool and excluded the isolates assigned to a sublineage different from the L4. A recent research paper [14] presents several Mtb L4 genomes from Ecuador. This genome dataset is not available in the SRA database but was requested by the authors, and they kindly shared the raw reads with us. A flow chart that depicts the genome selection strategy can be found in the supplementary material (S1 Fig).

## Latin American L4 *Mycobacterium tuberculosis* genome analysis

We performed the same strategy for all the isolates starting with WGS paired-end reads. First, we filtered low-quality reads using CUTADAPT v2.10 [15]. Read ends with bases below the Q30 quality threshold were removed, then reads with lengths below 70 bases were excluded. Singleton reads were also excluded from further analysis.

Filtered reads were assembled using SPADES v3.14.1 [16]. The assemblies' descriptive statistics were calculated with an in-house python script.

Average sequencing depth was calculated using SAMTOOLS [17] coverage tool while the alternate allele count was obtained by counting the number of variants from the VCF file created with the program BCFTOOLS mpileup [18].

## Phylogenomic analysis

We used 2,726 *M. tuberculosis* conserved single-copy genes for the phylogenomic analysis. Individual genes were searched within each assembly using BLASTN [19]. Then, they were extracted and aligned each CDS with its respective homologues using MAFFT [20]. Then, aligned individual CDSs were merged using the program *catsequences* (https://github.com/ChrisCreevey/catsequences). As outgroups, *M. tuberculosis* sublineage L2 genomes T67 and T85 were included.

A Maximum-likelihood (ML) tree was computed using IQTREE2 v. 2.1.32 [21]. The matrix was organized into partitions with different substitutions models selected according to BIC

(Bayesian Information Criterion) [22]. The matrix included 540 taxa, including references and outgroups (L2), with 56 partitions, 2,895,445 total sites, and 8,339 parsimony-informative sites. One thousand pseudoreplicates of SH-aLRT and ultra-fast bootstrap were performed, and consensus tree generated was edited in FigTree v1.4.4 (http://tree.bio.ed.ac.uk/software/figtree/). The tree terminals that exhibited too long branches were removed to reduce noise in the phylogeny.

### Genome deletion analysis

To determine structural deletions, each genome was aligned to H37Rv using the MUMMER v2 DNADIFF [23] script and the ASSEMBLYTICS tool [24]. The deletion events list (coordinates and size) of each genome was obtained using an in-house python script that processed the assemblytics' output files. Using the H37Rv annotation we mapped each deletion coordinates to the gene or genes that affect and plotted the deletion frequency on each lineage using *matplotlib*. Figures of selected deletions were prepared using the program Artemis [25] and using as reference the *M. tuberculosis* H37Rv genome downloaded from GenBank, accession GCF_000195955.2.

### Statistical and graphical analysis

Statistical and graphical analyses like boxplots were performed in the R environment unless stated otherwise (R version 4.0.4 64, x86_64-apple-darwin17.0 (64-bit)) and R studio (Version 1.2.1335). The scaffold and alternate allele count quality thresholds were calculated in R using the box plot upper limit formula Q3 + 1.5 x IQR.

### Availability of data and materials

Raw reads were deposited at the NCBI SRA database under the bioproject PRJNA867148. There is also a reviewer link that will be active during the manuscript review process: https://dataview.ncbi.nlm.nih.gov/object/PRJNA867148?reviewer=okjkdkaevi75kfu8phla1a30su.

## Results

### Genome quality assessment of the Latin American *M. tuberculosis* L4 isolates

We searched the NCBI SRA database for Illumina WGS data of Latin-American *Mtb* isolates of the lineage 4 (L4), the most prevalent lineage in the Latin American region. We found WGS reads of public projects of *Mtb* isolates from Mexico, Guatemala, Costa Rica, Panama, Colombia, Peru, Brazil, and Argentina. We also included six *Mtb* L4 genomes from Ecuador whose reads are unavailable in the SRA database. In the case of Colombia, we generated 88 new genomes from infected patients that live in the Andean city of Medellin. In the case of countries like Argentina, Brazil, Peru, and Mexico, hundreds of genome projects are publicly available. For the rest, a limited number of genome projects are available, in some cases below 10.

The first analysis was directed to detect and remove genome projects that exhibited poor-quality data based on filtering of low-quality reads or poor assembly performance. With this goal, reads were filtered with a Q30 threshold at both ends, and reads with lengths below 70 bases, were filtered out. Singleton reads were also excluded. In Fig 1, boxplots depict the general assembly statistics of the scaffolds, including scaffold average read coverage (Fig 1, panel A), total assembled bases (Fig 1, panel C), N50 value (Fig 1, panel D), largest assembled scaffold (Fig 1, panel E), and scaffold count (Fig 1, panel F). Complementary, we analyzed the presence of alternate alleles using a self-mapping approach and a subsequent alternate allele

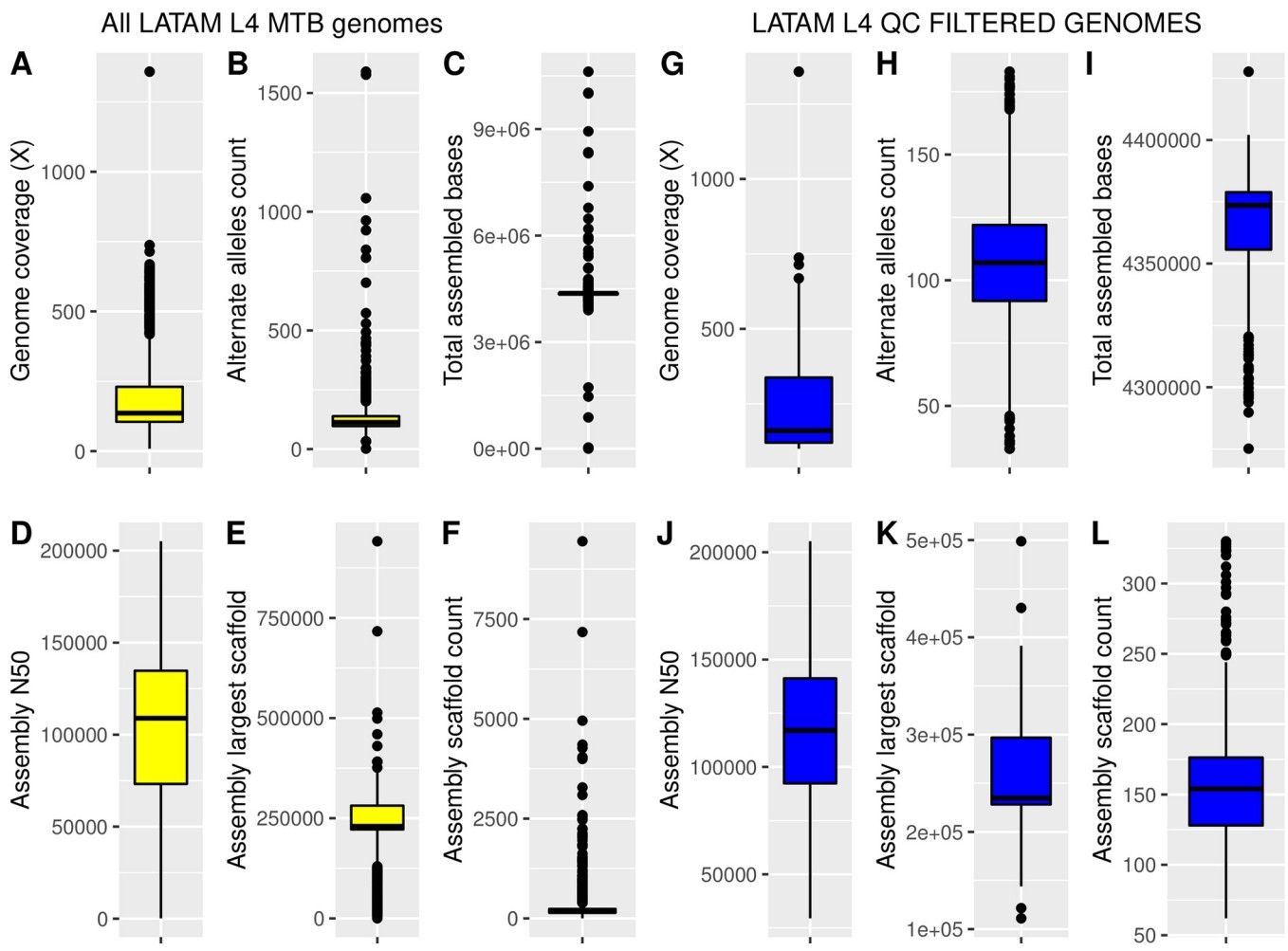

**Fig 1. Assembly statistics.** Boxplots depict the general assembly statistics of the scaffolds. Yellow boxes show the results of the initial 822 genomes analyzed (Panels A to F). Yellow boxes depict the results for the QC filtered 522 genomes (Panels G to L). Panels A/G, scaffold average read coverage (X). Panels B/H, alternate allele count. Panels C/I, total assembled bases. Panels D/J, N50 value. Panels E/K, largest assembled scaffold. Panels F/L, and scaffold count. The black line inside the boxes is the median value. Dots represent outliers.

calling and counting for each isolate (Fig 1, panel B). Since *Mtb* genome is haploid, detecting of any alternate allele indicates a mixture of genomes. Low alternate allele counts are normal even in clonal strains, but a more significant number might indicate a mixture of distant strains.

As shown in the Fig 1 boxplots, the assembly metrics showed a wide dispersion in all tested variables, especially in the sequencing coverage and assembly N50 measurements. The average sequencing depth ranged between 8.89X and 1358X with a median value of 136X. The N50 value showed a minimum value of 188 and a maximum of 205,097. The median N50 value was 108,926 bp. By contrast, the total assembled bases and scaffold count showed the narrowest dispersion ranges. Percentiles 25th and 75th for the total assembled bases were 4,348,965 and 4,380,025 bp., with a median value of 4,373,122 bp. In the case of the scaffold count, percentiles 25th and 75th were 141 and 238, with a median of 168 scaffolds. Outlier values observed in all tested variables indicated the presence of low-quality assemblies; for instance, five assemblies were below 2 Mb, 18 were above 5 Mb, and three were above 10Mb (indicating possible severe contamination in these 3 genomes). On the other hand, assembly fragmentation could also

indicate noisy genomic reads that lead to less accurate scaffold models. The median value of 168 scaffolds per assembly coincides with the expected performance for short-read technology. This scaffold count result agrees even with the old 454 WGS strategy. The narrow quartile ranges Q1 and Q3, 141 and 238, respectively, in such a large number of analyzed genomes, might denote what could be expected in good performing WGS experiment for this bacterium. According to the boxplot analysis, assemblies with more than 332 scaffolds are considered upper limit outliers. Notably, 43 assemblies showed to be highly fragmented with more than 1,000 scaffolds. One final metric allowed us to detect possible genome mixtures in the WGS experiments: the presence of high alternate allele counts. The median value for this metric was 111 alternate alleles in the tested genomes, corresponding to 0.0025% of the median genome assembly size. One conclusion that can be drawn from this result is that most genomes are highly clonal, and the expected accuracy of the scaffolds models presented in this work should be very high for most Latin American *Mtb* L4 genome assemblies.

Integrating these results, we can infer that, notwithstanding most of the genomes have acceptable quality assembly metrics, several genomes denote poor quality and must be removed to reduce noise in subsequent phylogenetic and comparative genomic analysis. In this sense, we decided to perform low-quality genome filtering. To do so, we focused on three different thresholds: i) average sequencing depth, ii) scaffold count, and iii) alternate allele counts. Combining these three thresholds allow the removal of low-covered, highly frag-mented, or mixed genomes. The average sequencing depth threshold was set to 100X, as is rec-ommended for most Illumina short read WGS experiments. The scaffold and alternate allele count thresholds were set using the box plot upper limit formula Q3 + 1.5 x IQR. In the case of the scaffold count, the threshold was set at 332, while for the alternate alleles, it was set at 186. Using these three filters, out of the 866 isolates that we included at the beginning, only 522 were retained for phylogenetic and comparative genomic analysis, as follows: Argentina (n = 210), Brazil (n = 86), Colombia (n = 88), Ecuador (n = 6), Guatemala (n = 10), Mexico (n = 10), and Peru (n = 112). In Fig 1, panels G to H, we present boxplots that depict the fil-tered assembly genome metrics.

## Phylogenomic analysis of the *M. tuberculosis* Latin-American L4 lineage

We wanted to gain insights into the sublineage assignation of the 522 filtered Latin American *Mtb* L4 genomes. Using the *TB-profiler* tool, we found that L4.1.1, L4.3.2, and L.4.8 are the most widespread sublineages being present in at least five different Latin American countries (Fig 2). However, the L4.1.2.1 was the most prevalent lineage accounting for nearly 47% of the filtered genomes. In conclusion, the L4.1 is the more successful sublineage in Latin America, representing around 57% of the Mtb analyzed (filtered) genomes. Peru showed the most diverse population of Mtb genomes with 10 different sublineages, and Argentina was the less diverse with just 2 sublineages (Fig 2). In a few cases, the *TB-profiler* tool reported two different L4 sublineages for one single isolate.

A maximum-likelihood phylogenomic tree was constructed using 2,726 parsimony infor-mative single copy genes to confirm and complement the barcoding TB-profiler classification of the 522 genomes with an evolutionary perspective (Fig 3 and S2 Fig). The phylogenomic tree recreates the already described topology of the *Mtb* L4 sublineages, with 100% UF-boot-strap support for all basal nodes. At the tips of the tree, the lowest branch support was 95, sug-gesting good confidence in the evolutionary history represented in the tree for the different Latin-American isolates. Notably, we can see that the L4 lineage is divided into two major clades. The first one contains only the L4.1 sublineage and the second encompasses the remaining sublineages (4.3 to 4.9). Furthermore, the tree also shows that the sublineages

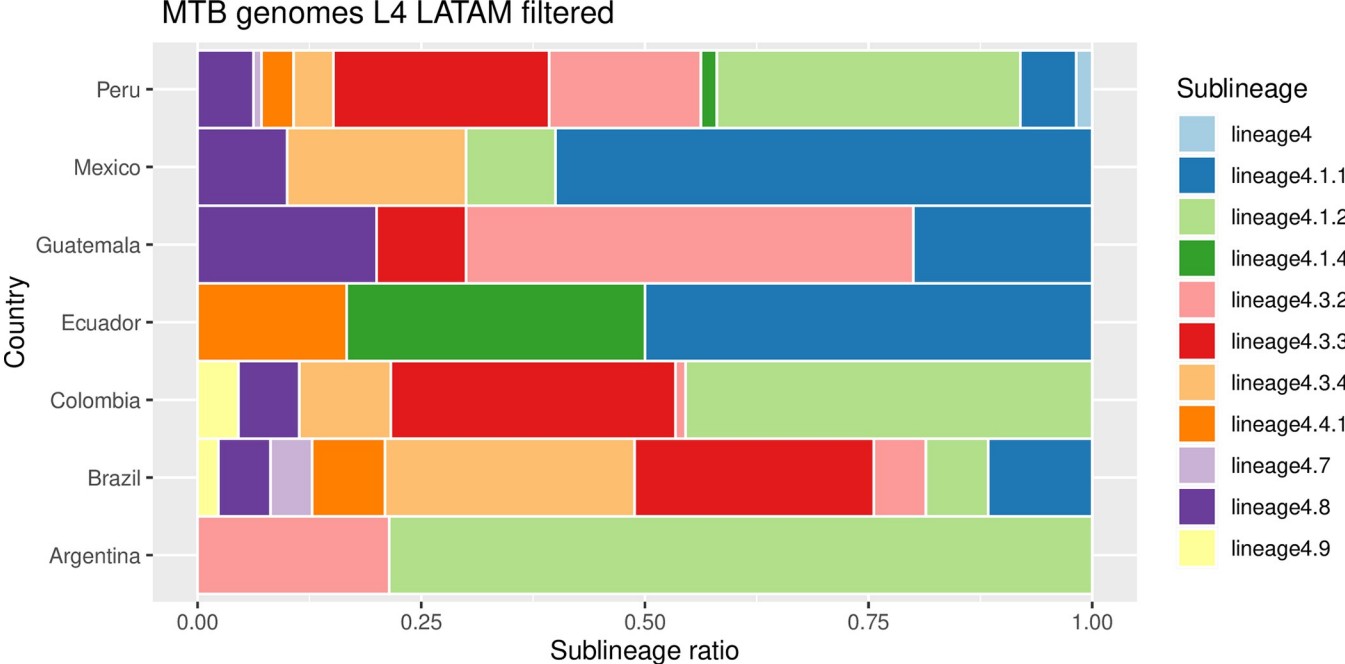

**Fig 2. Stacked bar plot representing the *TB-profiler* sublineage classifications within each country.** Argentina (n = 210), Brazil (n = 86), Colombia (n = 88), Ecuador (n = 6), Guatemala (n = 10), Mexico (n = 10), and Peru (n = 112).

4.1.1.1, 4.3.2, and 4.8, show the separation into two respective lineages that have not been reported for this subcontinent. Interestingly, while the sublineages from 4.3 to 4.9 show a more widespread distribution in south American countries, for instance, Colombia, Brazil, and Peru, three more modern lineages of the 4.1.2.1 sublineage show a well-supported country-specific pattern for countries like Colombia, Peru, and Argentina: 4.1.2.1Col1, 4.1.2.1Peru1, and 4.1.2.1.1Arg, respectively (Fig 3).

### Latin American L4 *M. tuberculosis* drug resistance profiles

Using the TBprofiler tool, we could detect gene mutations that render the bacteria resistant to commonly used antibiotics (S3 Fig). We found a wide distribution of antibiotic-resistant strains in all lineages and sublineages. We can observe the presence of Mtb isoniazid-resistant strains (Hr-TB), rifampicin-resistant strains (RR-TB), multi-drug resistant strains (MDR-TB), pre-Extensively Drug Resistant (pre-XDR-TB), and sensitive strains. Nevertheless, the majority of Mtb strains are sensitive across all sublineages.

### Genome deletions analysis in Latin American L4 *M. tuberculosis* strains

Genomic deletions are common in *M. tuberculosis* genomes. To gain insights into the genome deletions profiles of the L4 Latin American isolates, considering the advantage of having the genomes assemblies, we focused on large deletion events, i.e., >30 bases. Large deletions are usually overlooked using typical read mapping strategies. Compared to the H37Rv genome reference, the genomic deletions observed in the 522 filtered genomes ranged between 1 and 10,039 bases, being, as already reported, the most common ones those involving only 1 base (n = 44,366) (Fig 4) [7]. Large deletions (≥ 30 bases) were also common, with a total count of 23,882 events. It is noteworthy to mention that we also observed 4,111 deletion events that exceeded 1,000 lost bases.

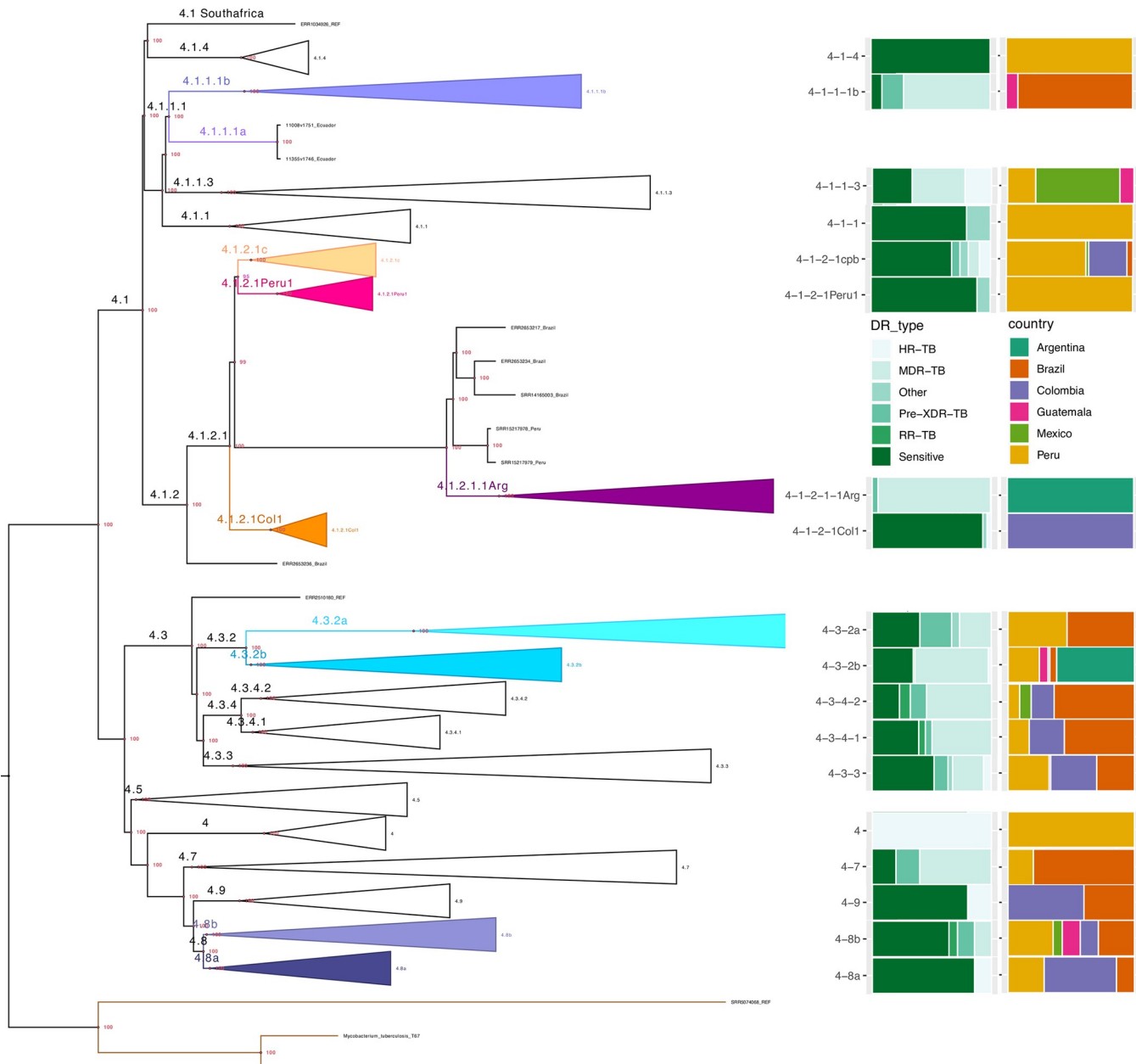

**Fig 3. Phylogenomic tree.** Maximum-likelihood phylogenomic tree constructed using 2,726 parsimony informative single copy genes tree depicting the phylogenetic relationships among the *M. tuberculosis* Latin-American L4 QC filtered genomes. At the bottom, in brown lines, is the L2 outgroup. The terminals were collapsed to reduce the size of the figure. The colored collapsed branches indicate new lineages. Green stacked bars at the center represent the antibiotic resistance profile based on the *TB-profiler* tool. At the leftmost end of the figure, the colored stacked bars represent the frequency of each lineage within Latin-American countries.

Next, we wanted to study if there is any specific signal of the deletion profiles within the Latin American L4 lineages. To do so, boxplot analysis of the deletion events count, the accumulated lost bases per isolate, and the largest deletion event per isolate were performed, grouping the genomes according to their respective evolutionary lineage (Fig 5). The number of deletions events detected per isolate varied from 9 to 142, showing some lineages with higher deletion event counts, like 4.1.1.1b and 4.1.2.1col1. By contrast, sublineage 4.9 accumulated

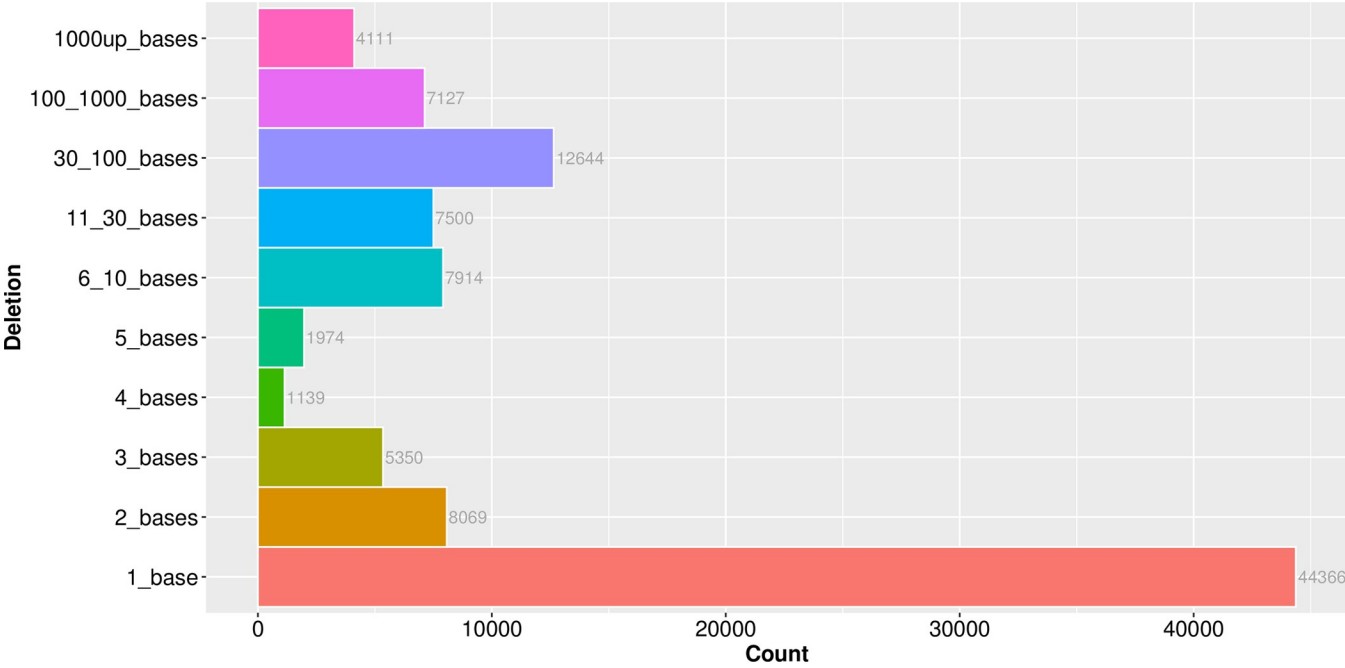

**Fig 4. Genomic deletions.** Bar plot depicting the frequency of the genomic deletions found on the 522 *M. tuberculosis* L4 QC filtered genomes grouped by their length in bases.

fewer deletions events with a median of 41. We calculated the total lost bases by adding the individual lengths of each deletion event. This analysis allowed us to see how many bases can be lost within each lineage. The sublineage 4.9 showed the less lost genome value with a median of 585 bases, while the 4.1.1.1b lineage showed the highest genome losses with a median of 10,875 lost bases. Again, as observed in the previous variables, total genome loss showed different patterns depending on the analyzed lineage. Lastly, the largest deletion event was spotted and compared among the Mtb L4 Latin American lineages. This boxplot analysis allowed us to differentiate, based on the median size of the largest genome deletion, several lineages or groups of related lineages that specifically share a particular deletion; like 4.1.2.1 (4.1.2.1.1Arg, 4.1.2.1Col1, 4.1.2.1cpb, and 4.1.2.1Peru1), 4.3.2a/b, 4.3.3 and 4.3.4 (4.3.4.1 and 4.3.4.2) (Fig 5). Remarkably, a large deletion of 6,479 bases appears like a signature for all 4.1.2.1 strains. This deletion affects 10 contiguous genes.

Given the importance of gene deletions as a source of loss of gene function, we wanted to annotate and compare, from an evolutionary perspective, the genes affected by these large deletions. Thus, we positioned the coordinates of the large genomic deletion on the reference chromosome of *Mtb* H37Rv and spotted the affected genes. We performed this analysis by grouping the deletion events according to the respective L4 lineages and we also quantified the relative frequency of each deletion event per lineage in the form of a heatmap (Fig 6). Most of the deletion events affected one single gene, and according to the genomic coordinates of the affected genes, these losses occurred along the whole bacterial chromosome. Conversely, the largest deletion events sometimes spanned several genes. Nucleotide losses in PPE and PE_PGRS genes were the most common.

While some of the genes affected by deletions showed patterns related to the phylogenetic background of the isolates and looks like synapomorphies, other do not follow this scheme and appear to behave more likely as a polyphyletic character. In this sense, it is noteworthy to

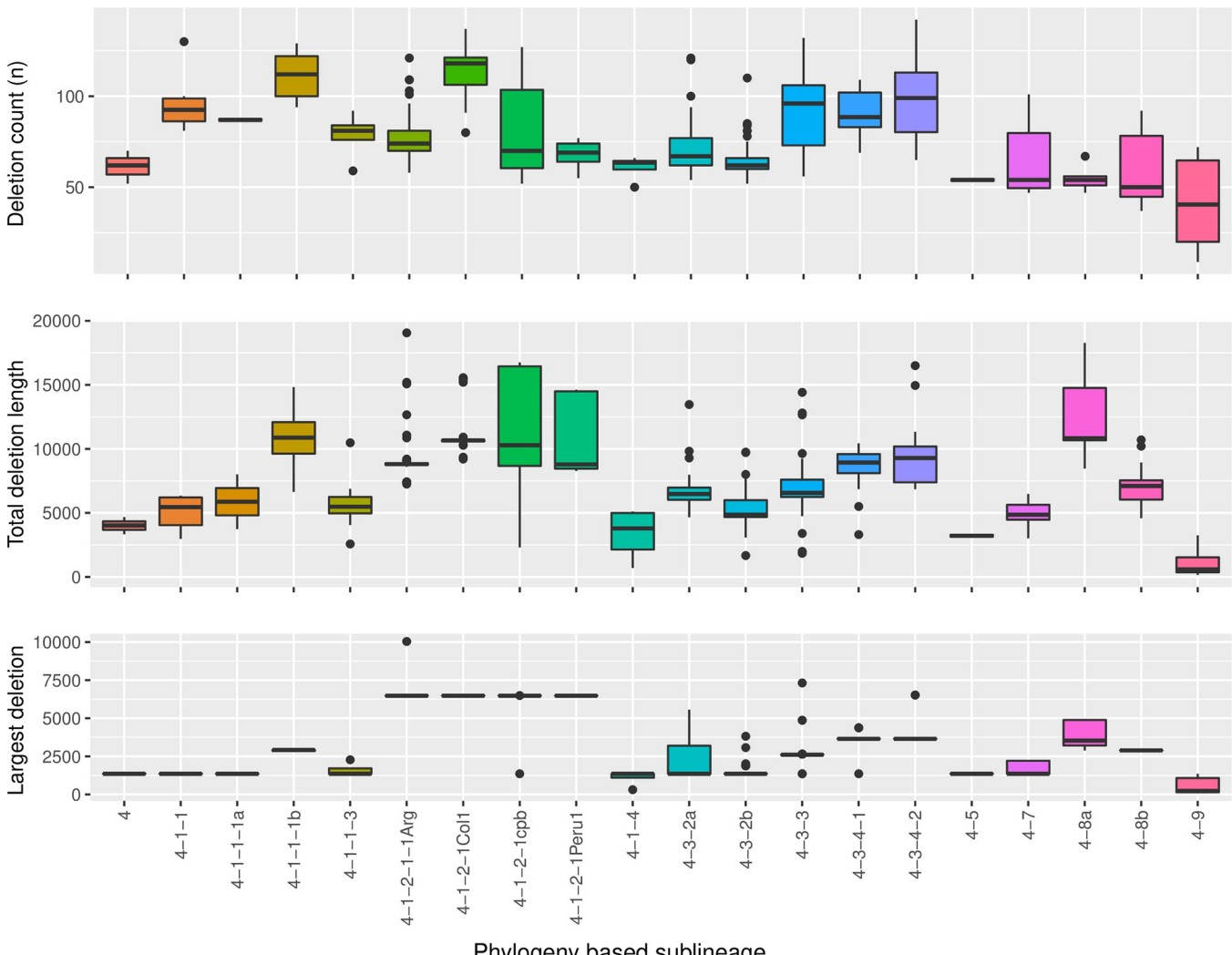

**Fig 5. Principal deletion events.** Boxplots depict quantitative characteristics of deletion events observed on the 522 Latin-American *M. tuberculosis* L4 QC filtered genomes grouped according to their evolutive lineage. The top panel presents the results of the sum of the deletion events, the middle panel presents the total accumulated lost bases per isolate, and the bottom panel shows the distribution of the largest deletion event in bp.

mention that genes like *ppe24*, *ppe8*, *rv2277c*, and *accE5*, accumulated mutations in most lineages, following a non-vertically driven inheritance pattern. By contrast, some deletion events served as signatures for some lineages. For instance, a single deletion event that spans genes *rv3083*, lipR, and *rv3085* shows a trend as a marker for the 4.8 sublineage in the Latin American *M. tuberculosis* genomes. On the other hand, a deletion that spans over the genes *eccD2*, *rv3888c*, and *espG2* seem specific for the lineage 4.1.1.1b.

We detected the 3.6 kb genomic deletion known as RD174 that is specific of the lineage 4.3.4 and extends over the genes *ctpG*, *rv1993c*, *cmtR*, *rv1995*, *rv1996*, and *ctpF*.

We also found two novel deletion events that specifically spot lineage 4.1.2.1. One of these deletions affects gene PPE69, and the other is a larger deletion that spans 6,479 nucleotides and affects the genes *ippN*, *rv2271*, *rv2272*, *rv2273*, *mazF8*, *mazE8*, *rv2275*, *cyp121*, *rv2277c*, and *rv2280* (Fig 7, panel A). Additionally, several strains of the sister clades 4.1.2.1Col1, 4.1.2.1Peru1, and 4.1.2.1cpb share a specific large deletion that spans 4,752 bases and that affects genes *rv1353c*, *rv1354c*, *moeY*, and *rv1356c* (Fig 7, panel B). Finally, we observed

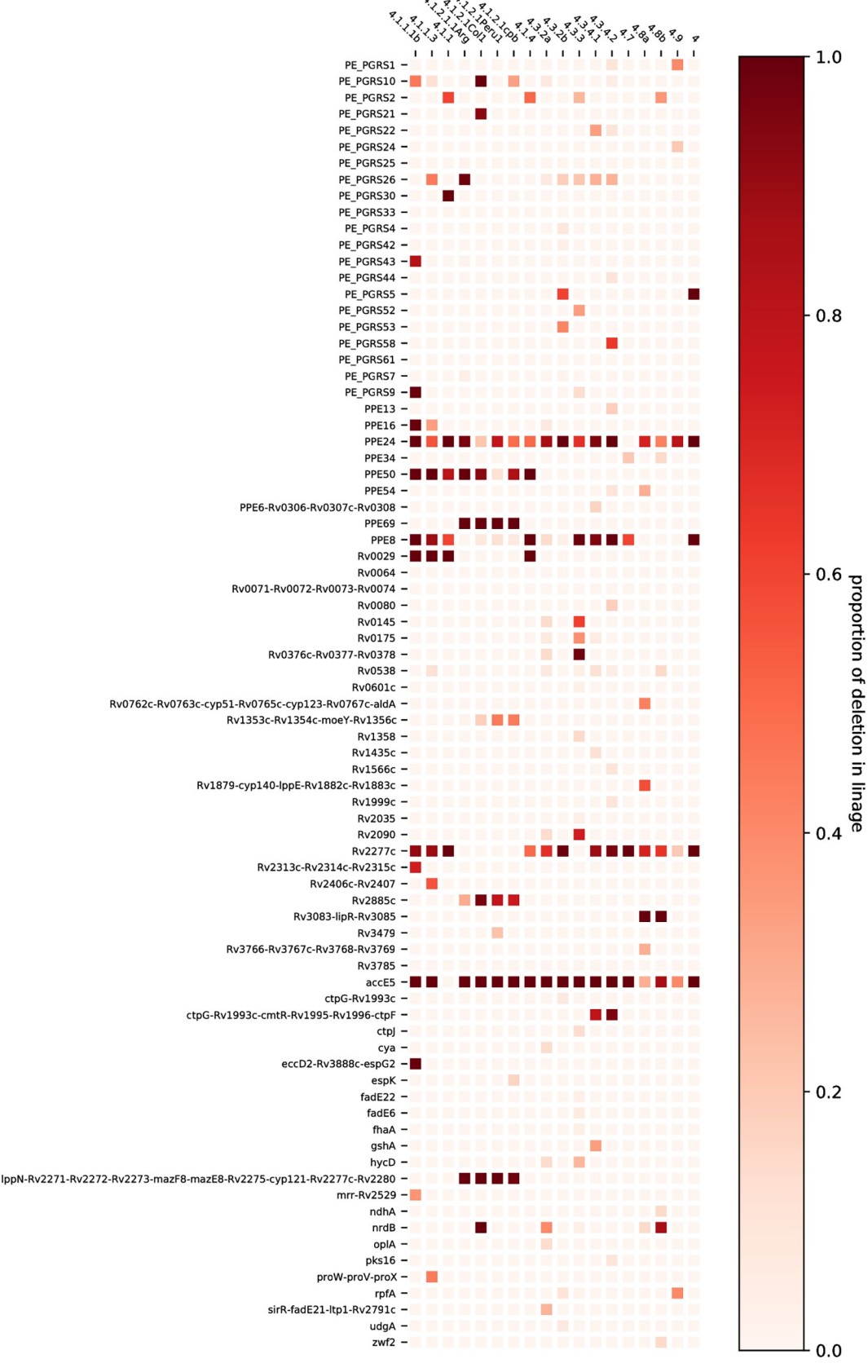

**Fig 6. Genes affected by deletions.** Genes affected by deletions are ranked according to lineages. The lineages are shown on the x-axis, while the y-axis shows the genes. The intensity of the color represents the frequency of the deletion in a particular gene within each lineage (the proportion of genomes that harbor a deletion in the gene). When one deletion affects several genes, the list of all affected genes is written on the y-axis.

another novel deletion that affects several strains in the lineage 4.8.a. This deletion spans 4,886 base pairs and affects genes *rv0762c*, *rv0763c*, *cyp51*, *rv0765c*, *cyp123*, *rv0767c*, and *aldA* (Fig 7, panel C).

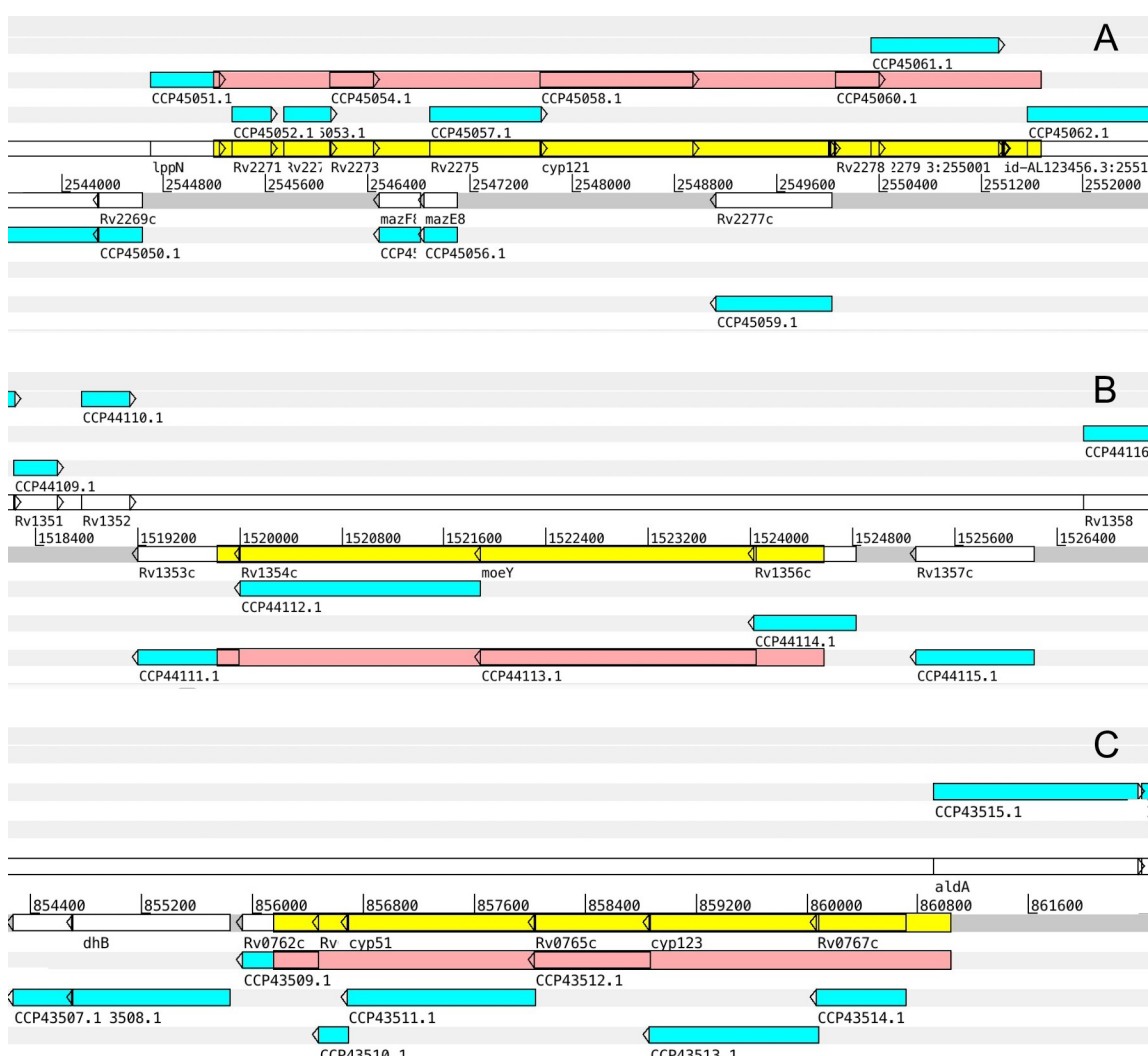

**Fig 7. Graphical representation of three selected (novel) genomic deletions observed within the Latin American L4 MTB genomes.** The top half of each figure depicts the plus DNA strand of the *M. tuberculosis* H37Rv reference genome with its nucleotide coordinates and its respective reading frames, while the bottom half depicts the minus strand similarly. Gene CDS sequences are depicted as white rectangles, while blue rectangles depict predicted peptide sequences. The deletion event on the chromosome is labeled as a yellow rectangle while the affected peptide regions are labeled with a pink rectangle. **Panel A**. Genomic deletion of 6,479 bp. that is specific to the Latin American L4.1.2.1 lineage and is present in 249 genomes. This deletion affects the genes *ippN*, *rv2271*, *rv2272*, *rv2273*, *mazF8*, *mazE8*, *rv2275*, *cyp121*, *rv2277c*, and *rv2280*. **Panel B**. Genomic deletion of 4,752 bp. that is common in the Latin American lineages 4.1.2.1Col1, 4.1.2.1Peru1, and 4.1.2.1cpb, and is present in 30 genomes. This deletion affects genes *rv1353c*, *rv1354c*, *moeY*, and *rv1356c*. **Panel C**. Genomic deletion of 4,886 bp. that is present in some genomes of the Latin American lineage 4.8.a and is present in 3 genomes. This deletion affects genes *rv0762c*, *rv0763c*, *cyp51*, *rv0765c*, *cyp123*, *rv0767c*, and *aldA*.

## Discussion

Current genomic technologies have pushed microbiology to a new era, and *M. tuberculosis* is probably one of the few bacterial models where the progress is more notable. In this sense, phylogenomic analysis unraveled that what we considered a single organism for nearly two centuries is, in fact, a complex mixture of nine different lineages with evident biological, geographical, and pathogenic dissimilarities [6, 26, 27].

Any reliable genomic analysis starts with good quality sequencing reads, but the typical read quality metrics based on PHRED scores alone appear insufficient to filter out poor quality genomic libraries. To tackle this situation, in the first part of this work, we wanted to assess the quality of Illumina shotgun reads of hundreds of Latin-American *Mtb* genomes deposited in the SRA database. We relied on commonly used descriptive statistics to compare the assembly performance and used a complementary strategy to detect a mixture of different *Mtb* strains. Despite that most genomes displayed a suitable profile of raw read quality metrics, our analysis noticed that nearly 40% of the read datasets were of poor quality and exhibited low coverage, contamination, or noisy DNA sequence signal. We advised researchers to perform a similar analysis to check *Mtb* read datasets used for *de novo* assembly, evolutionary or comparative genomic analysis. Our assembly metrics thresholds might be a good starting point for *Mtb* genome quality filtering.

*M. tuberculosis* lineage 4 showed to be diverse within the Latin-American countries, but some lineages showed to be widespread among several countries while others showed a narrower distribution [6]. We observed that *Mtb* sublineages L4.1.1, L4.1.2, L4.3.3, and L4.3.2 dominated in Latin-American human populations. However, we must stress that most of the strains included in this study were collected by other researchers, some in TB outbrake studies. This condition might introduce bias to the strain frequencies. Nevertheless, it must also be mentioned that we included a large number of strains that geographical cover from Mexico to Argentina, an area of nearly 20 million km$^2$.

Our high-resolution phylogenomic analysis included 2,726 CDSs that showed to be parsimony-informative, allowing us to portray a well-supported evolutionary history of the Mtb L4 in Latin America. The tree coincided with previous reports and showed a good concordance with the *TB-profiler* tool [4–6, 27, 28]. Nevertheless, the program reported two different, sometimes none-related, sublineages in a few strains. Our phylogenetic analyses positioned these conflictive samples in one lineage in the tree with 100% support. While the *TB-profiler* tool uses a small number of SNPs to classify the genomes (<100), our phylogenomic approach included 8,339 parsimony-informative sites.

The phylogeny also revealed that sublineages 4.8, 4.3.2, and 4.1.1.1 split into two well-supported internal clades. One additional observation is that sublineage 4.1.2.1 showed a more complex evolutionary past and wide distribution in most studied south American countries. This sublineage showed to be pretty successful and spreads throughout the whole subcontinent from Colombia (the northwesternmost nation in south America) to one of the most southern countries, Argentina. Furthermore, it shows several internal lineages and three of them with country-specific distributions: 4.1.2.1Col1 (Colombia), 4.1.2.1Peru1 (Peru), and 4.1.2.1.1Arg (Argentina). Similar observations, regarding geographical structure of this mycobacteria, have been observed in similar investigations performed in other *M. tuberculosis* lineages and continents [6, 29].

Regarding the resistance profiles of the Latin American L4 *Mtb* strains, it's clear that the antibiotic resistance genotype does not show a vertically inherited pattern associated with specific lineages. On the contrary, most sublineages exhibited a mixture of resistant genotypes. In

*Mtb* and other bacterial pathogens convergent evolution has been associated with the phenomenon of antibiotic resistance [5, 30, 31].

Genomic deletions are sculpting the *Mtb* L4 genomes of the future. Large deletions were observed in all Latin America L4 *Mtb* genomes but, interestingly, some lineages accumulated a higher number of deletions with their respective genome losses. Deletions spanned from 1 to 10kb, and it was common to find events where more than 1,000 bases were lost. Ten percent of the analyzed Latin American *Mtb* L4 strains have lost at least 0.25% of their genome. Most large deletions affected coding sequences, explained in part thanks to the high gene density of the Mtb genome [32]. Recent works on *Mtb* genomic deletions, saw congruent findings with our results within the *Mtb* lineages studied [33, 34], Nonetheless, as will be discussed below, our focus on Latin American genomes allowed us to detect novel large deletions that haven´t been reported so far.

The deletion events occurred along the whole chromosome, but as already described, there is a bias towards the repetitive gene families PE_PGRS and PPE [8, 10, 12]. Single base losses were the most common, and they are described as a reversible phenomenon that adds genetic plasticity to this pathogenic bacterium [7]. Nonetheless, large DNA deletions seem unlikely to be reversed and, in some cases, become genomic signatures for certain sublineages. Our evolutive genome analysis of the large deletions exposed that some of these events serve as signatures for several South America L4 sublineages, for instance, 4.1.2.1, 4.3.2, 4.3.3, and 4.3.4; these lineages show higher frequencies in the studied South American nations.

The genes *ppe24*, *ppe8*, *rv2277c*, and *accE5*, showed some bias to accumulate deletion regardless of their evolutive lineage. The PPE24 and PPE8 proteins belong to the multicopy family PPE, they encode 1,051 and 3,300 amino acids proteins, respectively, and their function is still unknown. Gene *rv2277c* encodes for a 301-aminoacids putative glycerolphosphodiesterase of unknown function. The gene *accE5* encodes for a probable bifunctional acetyl-/propionyl-coenzyme A carboxylase (epsilon chain) (177 amino acids) involved in synthesizing long-chain fatty acids.

Sublineage 4.8 signature deletion affects the genes *rv3083*, *lipR*, and r*v3085*, which products are annotated as FAD-containing monooxygenase MymA, acetyl-hydrolase LipR, and oxido-reductase SadH, respectively. Sublineage 4.1.1.1b signature deletion affects two proteins related to the ESX-2 type VII secretion system and a probable membrane protein.

In the case of lineage 4.3.4, the 3,650 bases deletion is a signature scar in its genome.

We previously reported this deletion in the year 2012 in a Colombian Mtb strain (UT205 strain) that extends over the genes *ctpG*, *rv1993c*, *cmtR*, *rv1995*, *rv1996*, and *ctpF*, part of the DosR regulon [12]. This specific deletion has been labelled as RD174. These results agree with the works recently published by Liu et al. and Bespiatykh et al. and confirms that this deletion is a signature of the 4.3.4 lineage [33, 34]. This deletion was inherited in the two branches of this lineage: 4.3.4.1 and 4.3.4.2. It is noteworthy that this deletion differentiates 4.3.4 from its sister clade 4.3.3. These two lineages have been demonstrated to have biological and virulence disparities [27].

One of the most striking lineage-specific deletions is the one that spans 6.5 kb, which is specific to the 4.1.2.1 lineage. Despite previous efforts oriented to describe MTB large genome deletions, this deletion was overlooked in the works published by Bespiatykh et al. [34] and Liu et al. [33] and is reported for the first time in this paper. This deletion affects a complex group of 10 genes positioned at both DNA strands. At the 5' end of the deletion, within the sense strand, organized in one operon-like structure, affects four genes annotated as lipoprotein LppN, hypothetical protein, and two transmembrane proteins. Subsequently, in the antisense strand, the toxin-antitoxin (TA) operon *mazE8*/ *mazF8* (Pandey and Gerdes, 2005) is completely lost, as well as two neighbor genes that partially overlap *mazE8*: *rv2275* and *cyp121*.

These former two genes encode for a probable cyclo(L-tyrosyl-L-tyrosyl) synthase, and a protein annotated as cytochrome P450, respectively. This Toxin-antitoxin locus is usually lost is host-associated bacteria [35]. The following lost gene is located at the antisense strand, *rv2277c*, which encodes a glycerolphosphodiesterase. Lastly, this large deletion ends at its 3' in a DNA repetitive element that is annotated as an IS6110 transposase. All the genes described above, that are affected by the large deletions, have been described as non-essential for *in vitro* growth [36, 37].

Another two interesting novel deletions that we observed in the Latin American genomes were those that were partially shared by its respective sublineages 4.8a or 4.1.2.1Col1, 4.1.2.1Peru1, and 4.1.2.1cpb. The former (present in sublineage 4.8a) disrupts the genes *rv0762c*, *rv0763c*, *cyp51*, *rv0765c*, *cyp123*, *rv0767c*, and *aldA*. Rv0762 and Rv0767 are annotated as conserved hypothetical proteins. Rv0763 is annotated as a Possible ferredoxin. Cyp51 and Cyp123 are annotated as non-essential P450 Cytochromes. Rv0765 is annotated as a non-essential probable oxidoreductase. Finally, AldA is annotated as a probable NAD-dependent aldehyde dehydrogenase that is also non-essential. The latter disrupts partially the genes *rv1353c* and *rv1356c* and entirely the genes *rv1354c* and *moeY*. Rv1354c and Rv1356c are annotated as hypothetical proteins, while Rv1353c and MoeY are annotated as Probable transcriptional regulatory proteins and Possible molybdopterin biosynthesis proteins, respectively. All these last genes are considered non-essential after transposon mutagenesis experiments [36].

Despite the broad spectrum of deletions that we observed in the L4 *Mtb* lineage infecting Latin-American individuals, it's clear that the bacteria are not losing their capacity to thrive in humans notwithstanding their gene losses. Gene loss, even loss of biochemical pathways, is an adaptation mechanism for specialist intracellular pathogens. This phenomenon was demonstrated in protozoan parasites like *Cryptosporidium* spp. [38, 39] and in pathogenic bacteria like *Shigella* [40]. The latter evolved from *E. coli* showing an accelerated process of gene loss and acquiring plasmids that harbor virulence genes. This process rendered the bacteria highly specialized in the human intestine and highly virulent [41–43]. For *Mtb* L4 Latin American strains we might witness a similar process where the gene losses shape the genetic adaptations of these mycobacteria to the changing Latin American human populations.

## Supporting information

**S1 Fig. Flow chart depicting the genome selection strategy.**
(TIF)

**S2 Fig. Maximum-likelihood phylogenomic tree constructed using 2,726 parsimony informative single copy genes tree depicting the phylogenetic relationships among the *M. tuberculosis* Latin-American L4 522 QC filtered genomes.** At the bottom, in brown lines, is the L2 outgroup. Numbers at nodes indicate ultrafast bootstrap support. Sublineages are labeled in their respective branches and highlighted with different colors.
(TIF)

**S3 Fig. Stacked bar plot representing the drug-resistant type profiles of the different Latin American M. tuberculosis L4 sublineages.** Drug-resistant types are depicted as normalized values on the X axis.
(TIF)

**S1 File.**
(CSV)

## Author Contributions

**Conceptualization:** Andres Baena, Luis F. Barrera, Juan F. Alzate.

**Data curation:** Andres Baena, Felipe Cabarcas, Luis F. Barrera, Juan F. Alzate.

**Formal analysis:** Andres Baena, Felipe Cabarcas, Luis F. Barrera, Juan F. Alzate.

**Funding acquisition:** Andres Baena, Juan F. Alzate.

**Investigation:** Andres Baena, Felipe Cabarcas, Juan C. Ocampo, Luis F. Barrera, Juan F. Alzate.

**Methodology:** Juan C. Ocampo, Juan F. Alzate.

**Software:** Felipe Cabarcas, Juan F. Alzate.

**Validation:** Andres Baena, Juan F. Alzate.

**Visualization:** Juan F. Alzate.

**Writing – original draft:** Andres Baena, Juan F. Alzate.

**Writing – review & editing:** Andres Baena, Felipe Cabarcas, Luis F. Barrera, Juan F. Alzate.

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
