## [Decision Letter · Decision Letter 0]

9 Feb 2023

PONE-D-22-28017Large genomic deletions delineate Mycobacterium tuberculosis L4 sublineages in South American countriesPLOS ONE

Dear Dr. Juan Fernando Alzate,

Thank you for submitting your manuscript to PLOS ONE. After careful consideration, we feel that it has merit but does not fully meet PLOS ONE’s publication criteria as it currently stands. Therefore, we invite you to submit a revised version of the manuscript that addresses the points raised during the review process (point by point, as raised by reviewers). In particular, the paper appears too  lenghty and needs a major revision according to one of the two reviewers.May I add that, in an evolutionary genomic perspective, "South America" or "Latin America" does not make such cultural or geographical unity (indigenous and portuguese and spanish influences) and the paper  as such could be improved by tryin to distinguish specifically what is Peru-specific as compared to other south-american countries characteristics of MTBC.

We look forward to receiving your revised manuscript.

Kind regards,

Christophe Sola, Pharm.D. Ph.D.

Academic Editor

PLOS ONE

Journal Requirements:

Additional Editor Comments (if provided):

This paper deserves to be rewritten according to one of the two reviewers. the new findings are not evident. it is apparently legnthy, focus on the main discoveries and on the methodology if brand new

Reviewers' comments:

Reviewer's Responses to Questions

**Comments to the Author**

1. Is the manuscript technically sound, and do the data support the conclusions?

Reviewer #1: Yes

Reviewer #2: Yes

2. Has the statistical analysis been performed appropriately and rigorously? 

Reviewer #1: Yes

Reviewer #2: N/A

3. Have the authors made all data underlying the findings in their manuscript fully available?

Reviewer #1: Yes

Reviewer #2: Yes

4. Is the manuscript presented in an intelligible fashion and written in standard English?

Reviewer #1: No

Reviewer #2: Yes

5. Review Comments to the Author

Reviewer #1: COMMENTS to authors

1. Major critique:

Overall, the paper should be made concise and clear

Now it is very lengthy.

Interest/novelty is not very clear. To me, the more interest may be not in study of the phylogeny in South America, but rather bioinformatics approach.

Other comments

2. Short Title: Genomic deletions in LATAM M. tuberculosis L4 lineage

LATAM abbreviation seems uncommon to me.

3. Title Large genomic deletions delineate Mycobacterium tuberculosis L4 sublineages in South American countries

Very general and obvious. It may be applied to any world region or globally.

4. ABSTRACT

Poorly structured with very long introductory sentences (half of abstract) and unclear part of what was the objective and content of this study. In contrast, some non-essential sentences are included (“Initially, we performed careful quality control of public read datasets and applied several thresholds to filter out low-quality data”)

How these genomes are representative: 522 L4 Latin American Mtb genomes

5. “Using a genome de novo assembly strategy and phylogenomic methods, we spotted new south American lineages that have not been revealed yet.”

Do you mean you have found new large deletions?

Otherwise, assembly of genomes is not needed for Mtb phylogenetic analysis (that is based on genome-wide SNPs, this species is devoid of HGT).

6. “Additionally, we describe genomic deletion profiles of these strains from an evolutionary perspective and report Mycobacterium tuberculosis L4 sublineages signature-like gene deletions.”

Are these deletions are novel? And never reported?

See Bespiatykh, D., Bespyatykh, J., Mokrousov, I., Shitikov, E. (2021). A comprehensive map of mycobacterium tuberculosis complex regions of difference. mSphere. 6 (4), e0053521. doi: 10.1128/mSphere.00535-21 and references therein

INTRODUCTION

7. Very long (this is sci paper not MS or PhD thesis).

METHODS

8. Bacterial culture and DNA extraction for the Colombian genomes – THIS SECTION IS TOO DETAILED. A ref is enough

9. Latin American Mycobacterium tuberculosis L4 genomic data downloaded from the NCBI SRA –

This section needs flowchart how to genomes were searched for and elected or filtered out

10. L4 is very large and heterogeneous and I am not sure that 522 genomes are sufficient

11. Some details on lanes 170-177 seem too technical

12. Lane 181 “We used 2,726 M. tuberculosis conserved single-copy genes for the phylogenomic analysis.” –

Do you mean that you used all these genes, 2,895,445 total sites? Not only variant snps found through alignment of reads to reference genome H37Rv.

This is quite redundant. Concatenated fasta of only variant snps would suffice.

Reviewer #2: This is a very nice manuscript telling us about Mycobacterium tuberculosis L4 lineages in South American countries.

I miss the information on the possibility if the samples are nationwide or not. I assume it is not. It could be a subtype bias and the distribution of subtypes can slightly different.

In this study only shortread WGS has been performed. Just one reference based on long and shortread would have been a help to the analysis instead of H37Rv. But that is optional!

Figure 1 can be excluded and thses results could be contained in the manuscript.

I would appreciate the space/not space in the M&M section between numbers and units are the same.

6. PLOS authors have the option to publish the peer review history of their article (what does this mean?). If published, this will include your full peer review and any attached files.

Reviewer #1: No

Reviewer #2: **Yes: **Erik Michael Rasmussen

---

## [Author Response · Author response to Decision Letter 0]

16 Feb 2023

In particular, the paper appears too lenghty and needs a major revision according to one of the two reviewers.

May I add that, in an evolutionary genomic perspective, "South America" or "Latin America" does not make such cultural or geographical unity (indigenous and portuguese and spanish influences) and the paper as such could be improved by tryin to distinguish specifically what is Peru-specific as compared to other south-american countries characteristics of MTBC.

R/: Dear Editor, thanks for your kind assistance in handling the manuscript. We appreciate the time that you and the reviewers invested in the process.

Regarding your comment that Latin America is not a cultural or geographical unit, we respectfully disagree.

The term Latin America is of French origin and was coined during the 1860s and refers to the American countries where languages were Spanish and Portuguese, and to a lesser extent, French. This concept has been gradually accepted since that time. You can find it in the Encyclopedia Britannica (https://www.britannica.com/place/Latin-America).

Geographically speaking, Latin America is well defined, comprising the countries of the American continent from Mexico to the south end of it. Similar biogeographic features make us share some similar ecosystems.

Culturally we share two main languages, Spanish and Portuguese, and the creed is mostly Catholic, with a high rate of practitioners of this religion. From the genetic point of view, Latin American populations are known to be among the most mestizo in the world, given the extensive genetic mixing that occurred between Latin Europeans, Amerindians, and Africans. This topic was well reviewed by Adhikari et al in 2016. (https://doi.org/10.1016/j.gde.2016.09.003). 

The contemporary population of Latin America is the result of complex, recent demographic events including admixture, bottlenecking, and subsequent recent, rapid post-colonial population growth. These demographic processes can result in founder effects that can create the conditions necessary for disease variants to drift to detectable frequencies, which directly affect disease susceptibility, and is critically important because it may determine the pathogenic strains that could be circulating among these specific populations (https://doi.org/10.1016/j.gde.2018.07.006).

This particular admixed genetic background has a substantial contribution to phenotypes related to health and disease, as it was demonstrated by Norris et al, 2018 (doi: 10.1186/s12864-018-5195-7). 

From the point of view of infectious disease profiles, we share similar profiles with high rates of intestinal and extraintestinal parasitic infections. Precisely this is a point of great difference with Saxon America, the USA, and Canada. In these countries, human populations have different genetic backgrounds with almost null ethnic mixing. Furthermore, USA and Canada are developed nations and display different cultural and infectious disease profiles. Additionally, in recent years the migration of Latin American citizens within the countries of the region has increased dramatically, favoring the phenomena of dispersion and homogenization of pathogenic microorganisms. Therefore, we believe that it is appropriate to understand tuberculosis within the framework of Latin American countries, given its geographical relationship, cultural similarities, population dynamics, and the mestizo genetic base of its population. Drawing a parallel between Latin America and the Asian or European regions, in the former two, where cultural, linguistic, and religious dissimilarities are much greater, there is no objection to understanding these blocks as defined geographical and cultural groups.

Reviewers' comments:

Reviewer's Responses to Questions

Comments to the Author

1. Is the manuscript technically sound, and do the data support the conclusions?

Reviewer #1: Yes

Reviewer #2: Yes

R:/ Thanks for the review

2. Has the statistical analysis been performed appropriately and rigorously? 

Reviewer #1: Yes

Reviewer #2: N/A

R:/ Thanks for the review

3. Have the authors made all data underlying the findings in their manuscript fully available?

Reviewer #1: Yes

Reviewer #2: Yes

R:/ Thanks for the review

4. Is the manuscript presented in an intelligible fashion and written in standard English?

Reviewer #1: No

Reviewer #2: Yes

R:/ Thanks for the review. English was reviewed again.

5. Review Comments to the Author

Reviewer #1: COMMENTS to authors

1. Major critique:

Overall, the paper should be made concise and clear

Now it is very lengthy.

Interest/novelty is not very clear. To me, the more interest may be not in study of the phylogeny in South America, but rather bioinformatics approach.

R:/ Thanks for the review. The manuscript was modified as suggested by the reviewers. The introduction and methods sections were reduced. The abstract was modified making it more concise and making explicit the novelty of the work. We consider that the bioinformatic approach, as well as the comparative genomic results, are both interesting for the scientific community.

Other comments

2. Short Title: Genomic deletions in LATAM M. tuberculosis L4 lineage

LATAM abbreviation seems uncommon to me.

R:/ LATAM abbreviation was changed by Latin American in the entire document.

3. Title Large genomic deletions delineate Mycobacterium tuberculosis L4 sublineages in South American countries

Very general and obvious. It may be applied to any world region or globally.

R:/ We partially agree with the reviewer, but still, we consider the title is original and valid. If it can be applied to any world region of the whole planet, may be, but for now this is a “probable” speculation. Our results are solid, and we focused our effort on Latin America. Moreover, we observed at least three novel large deletions in one of the most frequent sublineages observed in the Mtb-infected individual in South America.

4. ABSTRACT

Poorly structured with very long introductory sentences (half of abstract) and unclear part of what was the objective and content of this study. In contrast, some non-essential sentences are included (“Initially, we performed careful quality control of public read datasets and applied several thresholds to filter out low-quality data”)

How these genomes are representative: 522 L4 Latin American Mtb genomes

R:/ The Abstract was modified reducing the introductory sentences and adding more information about the aim of the study and the most relevant findings.

We started this work with more than 1000 Latin American strains that have public genomic data (SRA repository) with an acceptable sequencing depth. After taxonomy (L4 lineage) and quality filtering these 522 were the ones that best quality with our selection thresholds. For us, it was a paramount goal to reduce noise that can be introduced by low-qual genomic reads. Although you can increase the number of strains, this will not change our main findings of novel country-specific sublineage/clades and the genomic deletion profiles.

5. “Using a genome de novo assembly strategy and phylogenomic methods, we spotted new south American lineages that have not been revealed yet.”

Do you mean you have found new large deletions?

Otherwise, assembly of genomes is not needed for Mtb phylogenetic analysis (that is based on genome-wide SNPs, this species is devoid of HGT).

R:/ We are referring to the lineage definition according to a phylogenetic point of view. It is synonymous with clade, in this case, a subspecies clade that evolved in Latin American countries from the original European introduced L4 Mtb strains.

6. “Additionally, we describe genomic deletion profiles of these strains from an evolutionary perspective and report Mycobacterium tuberculosis L4 sublineages signature-like gene deletions.”

Are these deletions are novel? And never reported?

See Bespiatykh, D., Bespyatykh, J., Mokrousov, I., Shitikov, E. (2021). A comprehensive map of mycobacterium tuberculosis complex regions of difference. mSphere. 6 (4), e0053521. doi: 10.1128/mSphere.00535-21 and references therein

R:/ Indeed we found at least three novel genomic deletions. We compared our results to those published by Bespiatykh et al. (doi:10.1128/mSphere.00535-21) and Liu et al. (doi:10.3389/fmicb.2022.984582), and they didn’t find these three large deletions that we spotted here.

INTRODUCTION

7. Very long (this is sci paper not MS or PhD thesis).

R:/ The introduction length was significantly reduced in the revised version of the manuscript.

METHODS

8. Bacterial culture and DNA extraction for the Colombian genomes – THIS SECTION IS TOO DETAILED. A ref is enough

R:/ We deleted the description of the procedure of DNA extraction and left to references where the protocols are described in more detail.

9. Latin American Mycobacterium tuberculosis L4 genomic data downloaded from the NCBI SRA –

This section needs flowchart how to genomes were searched for and elected or filtered out

R:/ A flow chart was prepared and added as supplementary material.

10. L4 is very large and heterogeneous and I am not sure that 522 genomes are sufficient

R:/ We agree with the reviewer that L4 is a vast lineage, but it was demonstrated which sublineages were introduced into America since the XVI century (DOI:10.1126/sciadv.aat5869 ) and our results are in concordance with these finds. We analyzed more than 500 high-quality genomes representing 9 countries separated by thousands of kilometers. We think that for actual scientific similar works, this is a representative number. The Phylogenomic results also support the phylogeographic findings presented in the paper.

Additionally, ss has been already described elsewhere (Nat Genet. 2016 December; 48(12): 1535–1543. doi:10.1038/ng.3704.). L4 possess geographically restricted sublineages. This phenomenon is congruent to our observations of evolutionary patterns observed in Latin American human populations.

11. Some details on lanes 170-177 seem too technical.

R:/ We agree with the reviewer. But as other journals have pointed us, it is important to describe details of the bioinformatic pipelines to allow other researchers to reproduce the results. Following this comment from the reviewer, we removed several details. The description in the methods section was modified as follows:

“Filtered reads were assembled using SPADES v3.14.1 [22]. The assemblies' descriptive statistics were calculated with an in-house python script.

Average sequencing depth was calculated using SAMTOOLS [23] coverage tool while the alternate allele count was obtained by counting the number of variants from the VCF file created with the program BCFTOOLS mpileup [24].”

12. Lane 181 “We used 2,726 M. tuberculosis conserved single-copy genes for the phylogenomic analysis.” –

Do you mean that you used all these genes, 2,895,445 total sites? Not only variant snps found through alignment of reads to reference genome H37Rv.

This is quite redundant. Concatenated fasta of only variant snps would suffice.

R:/ Indeed we used all those genes. We understand that a typical read mapping strategy is more commonly used in this kind of work thanks to the lower computation demand. In our case, we preferred to follow a different strategy disregarding the computational load of it. We aimed to extract each orthologous CDS sequence, the complete CDS in most strains, and then align them individually. Then, we concatenated all the 2726 single-copy orthologous genes into one single supermatrix that was fed into the IQTREE2 program. Although the matrix was pretty big, IQTREE2's first step is to mask conserved sites and only work with those positions that are phylogenetically informative. In this case, the complete matrix comprises 2,895,445 sites, but only 8,339 were described for IQTREE2 as parsimony informative. This step does not require much time on our servers. This information is described in the Phylogenomic analysis subheading, second paragraph of the methods section.

Reviewer #2: This is a very nice manuscript telling us about Mycobacterium tuberculosis L4 lineages in South American countries.

I miss the information on the possibility if the samples are nationwide or not. I assume it is not. It could be a subtype bias and the distribution of subtypes can slightly different.

In this study only shortread WGS has been performed. Just one reference based on long and shortread would have been a help to the analysis instead of H37Rv. But that is optional!

R:/ The genomic data that we have generated, according to our limited budget, was in the city of Medellin-Colombia, the second largest city in Colombia, which has a high incidence rate of TB in the country. Luckily, there were public genome sequences of L4 TB isolates for another Colombian Andean City, Manizales. These last genomes (those from Manizales) were sequenced by another researcher a few years ago. Medellin and Manizales cities are in the Andean region of the country with similar geographic features and are separated by around 227 km.

The coverage is not nationwide, so we cannot confirm the national subtype frequencies. To fulfill this goal, we will require a much greater budget due to the size of the country, which is like 2 times Spain, and a large number of inhabitants, nearly 50 million.

We agree that we cannot refer to the actual frequencies of the sublineages along the country and, we could have missed some other subtypes. But the main findings of our work are well supported, taking into account that we wanted to see the broader picture of the historical evolution and genome deletion profiles on a subcontinental scale.

Figure 1 can be excluded and thses results could be contained in the manuscript.

I would appreciate the space/not space in the M&M section between numbers and units are the same.

R:/ We consider that the quality control of genomic data is extremely important to settle the validity of the evolutionary y comparative genomic analysis. This information can be useful for other researchers that want to replicate our bioinformatic strategy. In this sense, if it’s possible, we would like to keep Figure 1 as one main figure in the manuscript.

---

## [Decision Letter · Decision Letter 1]

24 Apr 2023

Large genomic deletions delineate Mycobacterium tuberculosis L4 sublineages in South American countries

PONE-D-22-28017R1

Dear Dr. Juan Fernando Alzate,

We’re pleased to inform you that your manuscript has been judged scientifically suitable for publication and will be formally accepted for publication once it meets all outstanding technical requirements.

Kind regards,

Li Xing

Academic Editor

PLOS ONE

Additional Editor Comments (optional):

Reviewers' comments:

Reviewer's Responses to Questions

**Comments to the Author**

1. If the authors have adequately addressed your comments raised in a previous round of review and you feel that this manuscript is now acceptable for publication, you may indicate that here to bypass the “Comments to the Author” section, enter your conflict of interest statement in the “Confidential to Editor” section, and submit your "Accept" recommendation.

Reviewer #1: All comments have been addressed

2. Is the manuscript technically sound, and do the data support the conclusions?

Reviewer #1: Yes

3. Has the statistical analysis been performed appropriately and rigorously? 

Reviewer #1: Yes

4. Have the authors made all data underlying the findings in their manuscript fully available?

Reviewer #1: Yes

5. Is the manuscript presented in an intelligible fashion and written in standard English?

Reviewer #1: Yes

6. Review Comments to the Author

Reviewer #1: (No Response)

7. PLOS authors have the option to publish the peer review history of their article (what does this mean?). If published, this will include your full peer review and any attached files.

Reviewer #1: No

---

## [Editor Report · Acceptance letter]

10 May 2023

PONE-D-22-28017R1 

Large genomic deletions delineate *Mycobacterium tuberculosis* L4 sublineages in South American countries 

Dear Dr. Alzate:

I'm pleased to inform you that your manuscript has been deemed suitable for publication in PLOS ONE. Congratulations! Your manuscript is now with our production department. 

Kind regards, 

on behalf of

Professor Li Xing 

Academic Editor

PLOS ONE